# N-Glycans in Immortalized Mesenchymal Stromal Cell-Derived Extracellular Vesicles Are Critical for EV–Cell Interaction and Functional Activation of Endothelial Cells

**DOI:** 10.3390/ijms23179539

**Published:** 2022-08-23

**Authors:** Marta Clos-Sansalvador, Sergio G. Garcia, Miriam Morón-Font, Charles Williams, Niels-Christian Reichardt, Juan M. Falcón-Pérez, Antoni Bayes-Genis, Santiago Roura, Marcella Franquesa, Marta Monguió-Tortajada, Francesc E. Borràs

**Affiliations:** 1REMAR-IGTP Group, Germans Trias i Pujol Research Institute (IGTP) & Nephrology Department, University Hospital Germans Trias i Pujol (HUGTiP), Can Ruti Campus, 08916 Badalona, Spain; 2Department of Cell Biology, Physiology and Immunology, Universitat Autònoma de Barcelona (UAB), 08193 Bellaterra, Spain; 3CIC biomaGUNE, Glycotechnology Group, Basque Research and Technology Alliance (BRTA), 20014 San Sebastian, Spain; 4Exosomes Laboratory, Center for Cooperative Research in Biosciences (CIC bioGUNE), Basque Research and Technology Alliance (BRTA), 20850 Derio, Spain; 5Centro de Investigación Biomédica en Red de Bioingeniería, Biomateriales y Nanomedicina (CIBER-BBN), 20014 San Sebastian, Spain; 6Centro de Investigación Biomédica en Red de Enfermedades Hepáticas y Digestivas (CIBERehd), Instituto de Salud Carlos III, 41092 Madrid, Spain; 7Basque Foundation for Science (IKERBASQUE), 48009 Bilbao, Spain; 8ICREC Research Program, Health Science Research Institute Germans Trias i Pujol, Can Ruti Campus, 08916 Badalona, Spain; 9iCor, Cardiology Service, Germans Trias i Pujol University Hospital, 08916 Badalona, Spain; 10CIBERCV, Instituto de Salud Carlos III, 41092 Madrid, Spain; 11Department of Medicine, Universitat Autònoma de Barcelona (UAB), 08193 Barcelona, Spain; 12Faculty of Medicine, University of Vic-Central University of Catalonia (UVic-UCC), 08500 Vic, Spain; 13Department of Cell Biology, Physiology and Immunology, Universitat de Barcelona (UB), 08007 Barcelona, Spain

**Keywords:** mesenchymal stem/stromal cells, extracellular vesicles, exosomes, glycans, glycosylation, lectins

## Abstract

Mesenchymal stromal cell-derived extracellular vesicles (MSC-EV) are widely considered as a cell-free therapeutic alternative to MSC cell administration, due to their immunomodulatory and regenerative properties. However, the interaction mechanisms between EV and target cells are not fully understood. The surface glycans could be key players in EV–cell communication, being specific molecular recognition patterns that are still little explored. In this study, we focused on the role of N-glycosylation of MSC-EV as mediators of MSC-EV and endothelial cells’ interaction for subsequent EV uptake and the induction of cell migration and angiogenesis. For that, EV from immortalized Wharton’s Jelly MSC (iWJ-MSC-EV) were isolated by size exclusion chromatography (SEC) and treated with the glycosidase PNGase-F in order to remove wild-type N-glycans. Then, CFSE-labelled iWJ-MSC-EV were tested in the context of in vitro capture, agarose-spot migration and matrigel-based tube formation assays, using HUVEC. As a result, we found that the N-glycosylation in iWJ-MSC-EV is critical for interaction with HUVEC cells. iWJ-MSC-EV were captured by HUVEC, stimulating their tube-like formation ability and promoting their recruitment. Conversely, the removal of N-glycans through PNGase-F treatment reduced all of these functional activities induced by native iWJ-MSC-EV. Finally, comparative lectin arrays of iWJ-MSC-EV and PNGase-F-treated iWJ-MSC-EV found marked differences in the surface glycosylation pattern, particularly in N-acetylglucosamine, mannose, and fucose-binding lectins. Taken together, our results highlight the importance of N-glycans in MSC-EV to permit EV–cell interactions and associated functions.

## 1. Introduction

Mesenchymal stromal cell-derived extracellular vesicles (MSC-EV) have already demonstrated their efficient modulation of inflammation and the exertion of regenerative properties in a number of both preclinical models [1] and clinical trials [2]. However, the specific mechanism of action of MSC-EV remains unclear.

In this context, it is widely accepted that EV cargo (lipids, RNA and proteins) can be shuttled to the target cells by endocytosis (clathrin-dependent or caveolae-dependent) [3,4,5,6], phagocytosis [7], macropinocytosis [8] or by plasma/endosomal membrane fusion in acidic pH conditions, when EV and cells’ plasma membrane display the same fluidity [9]. Moreover, it has also been established that this transfer of vesicular content plays a crucial role in the intercellular communication in several biological processes, including angiogenesis [10,11], cell migration [12], inflammation regulation [13,14], bone formation [15] or interneuronal communication [16]. The transcriptomic, proteomic and lipidomic profiles of EV have been widely studied to find the molecules implicated in EV–cell interaction [17]. However, the specific mechanisms of EV recognition and their subsequent capture by target cells are still not fully elucidated.

The implication of glycans in important cellular mechanisms in non-EV-related studies, such as cell adhesion [18,19], self- and non-self-recognition [20,21], molecular trafficking and receptor activation, has turned glycans into a new focus of molecules in which to find the possible mechanisms of EV function and targeting. In this context, a few studies (reviewed in [22,23]) have already reported EV glycan profiling analysis or specific receptor–ligand interactions for EV uptake in a glycan-dependent manner. Most of them referred to cancer-derived EV, urine EV or plasma EV. Specifically, heparan sulfate proteoglycans (HSPGs) were implicated in the uptake of tumor-derived EV by CHO cells [4], and in the paracrine intercommunication of the hepatic stellate cells [24]. The sialic acid residues have also been involved in EV-uptake in vitro [25,26] and in vivo [27], and the depletion of these sialic acid residues has also been shown to modify the EV biodistribution in mice [28], highlighting the importance of protein and lipid glycosylation in EV targeting.

Thus, in the present study, we aimed to evaluate the implication of MSC-EV N-glycosylation in EV–endothelial cell interaction, EV-uptake and also, in specific MSC-EV described functions, such as endothelial cell recruitment and tube-like formation stimulation. Our results showed that the removal of N-glycosylation by PNGase-F treatment on MSC-EV affected both capture and functionality on HUVEC.

## 2. Results

### 2.1. Isolation and Characterization of iWJ-MSC-EV

The isolated iWJ-MSC-EV were screened for the MSC marker CD90, and the EV markers CD63 and CD9. The SEC elution profile of iWJ-MSC-EV, treated or not with PNGase-F (Figure 1A), showed the EV elution indistinctly from the EV treatment. The EV were detected in fractions 4–6, nicely separated from the protein, which eluted later, confirming the previously published results [29]. Nevertheless, we observed a decrease in the MFI in the expression of all of the three markers after the addition of PNGase-F, especially affecting CD90 (CD9: 33%, *p* < 0.05; CD63: 24% n.s; CD90: 75%, *p* < 0.0001; *n* = 3; Figure 1B), which could be indicative of N-glycosylation sensitive epitopes. The fractions containing iWJ-MSC-EV were pooled together and analyzed by cryo-electron microscopy, which confirmed that both iWJ-MSC-EV and iWJ-MSC-EV PNGase-F had a double membrane, nanosize and round shape (Figure 1C). The iWJ-MSC-EV and iWJ-MSC-EV PNGase-F were also screened for calnexin and ezrin markers. The calnexin dotblot revealed no important reticulum endoplasmic contamination in the EV samples, and ezrin was positive for both of the iWJ-MSC-EV types (Figure 1D).

### 2.2. N-Glycosylation Is Critical to Uptake iWJ-MSC-EV by HUVEC

To study the role of iWJ-MSC-EV’s N-glycosylation in the physiology of EV–endothelial cell interactions, we first assessed the EV uptake by flow cytometry. The incubation of HUVEC with CFSE-labelled iWJ-MSC-EV resulted in an increase in cell fluorescence, whilst HUVEC incubated with PNGase-F-treated EV showed a significant reduction in EV uptake of about 50% compared to iWJ-MSC-EV (*p* < 0.05; *n* = 4) (Figure 2A,B). The flow cytometry results were further confirmed by fluorescence microscopy. Our results showed specific green fluorescence (CFSE-EV) in HUVEC’s cytoplasm after the capture of the untreated EV. After quantification, the untreated EV uptake by HUVEC was significant in terms of the mean intensity per cell and the green area per cell after autofluorescence correction. Only the background fluorescence was observed in the control (no EV) and in the PNGase-F-treated EV, contrarily to the untreated EV, indicating that the capture was abrogated with the PNGase-F treatment of EV (*p* < 0.0001) (Figure 2C,D). These data suggest that the EV N-glycans mediate interactions with HUVEC.

### 2.3. PNGase-F Treatment Abrogates HUVEC Recruitment by iWJ-MSC-EV

The previous studies demonstrated the capacity of porcine MSC-EV to induce the cell recruitment of allogeneic MSC and endothelial progenitor cells [30]. We thus next studied the potential role of the surface N-glycans of EV to modulate the chemotactic response of the allogenic HUVEC towards agarose spots—containing both types of iWJ-MSC-EV.

The iWJ-MSC-EV-embedded agarose spots induced the cell recruitment of HUVEC compared to control PBS or PNGase-F treated EV-embedded agarose spots (Figure 3A,B), both in terms of migration distance (20% reduction; *p* < 0.05 HUVEC) and total area occupied by cells (45% reduction; *p* < 0.01). The migration levels in the agarose spots containing PNGase-F-treated EV decreased near to the negative control levels.

### 2.4. Tube-like Structures Promoted by iWJ-MSC-EV Are Dependent on N-Glycosylation

To further investigate the functional involvement of N-glycans on EV functions, the primordial angiogenic capacity of HUVEC was measured in vitro. The microscopy images showed that, compared to the negative controls, iWJ-MSC-EV enhanced tube formation at 6 h of incubation in a dose-dependent manner, and to a comparable extent as the positive control: VEGF-A 10 ng/mL (VEGF) (Figure 3C,D). The PNGase-F-treated EV failed to induce tube formation, suggesting that N-glycosylation is also playing a role in this function. Both the number of the nodes and the total branching length were significantly higher when iWJ-MSC-EV was added, compared to the PNGase-F-treated EV (Figure 3D). This reduction particularly affected the number of nodes (85%), and, to a lesser extent, the total branching length (70%).

### 2.5. iWJ-MSC-EV and PNGase-F-Treated iWJ-MSC-EV Exhibit Distinct Surface Glycosylation Profile Patterns

Given the implied importance of N-glycosylation for iWJ-MSC-EV in these functional assays, a lectin microarray analysis comprising of 26 lectins was set up to discover which type of glycosylation features were present in iWJ-MSC-EV, and which were affected by the PNGase-F treatment. The iWJ-MSC-EV showed high fluorescent signals for N-acetylglucosamine (GlcNAc) binding the lectins wheat germ agglutinin (WGA), *Allium sativum* (ASA), *Solanum tuberosum* (STL) and *Ricinus communis* (RCA); and for the N-acetylgalactosamine (GalNAc) binding the lectin *Cratylia floribunda* (CFL). A lower binding was observed for the fucose-binding lectins *Pisum sativum* (PSA) and *Lens culinaris* (LCA); the mannose-binding lectin *Polygonatum multiflorum* (PMA); and a very low presence of glycans containing sialic acids or O-glycans (T-antigen). The glycans containing α-galactose residues were absent (Figure 4A). In contrast, the PNGase-F-treated EV showed a decreased binding signal for some of these lectins. In particular, a reduction was observed in the GlcNAc-binding lectins (ASA and RCA), in the mannose-binding lectin (PMA) and in the fucose-binding lectins (AAL, PSA and LCA) (Figure 4C). This strong decrease in the signal after PNGase-F treatment suggests that most of these glycosidic residues reside on N-glycans and that the enzymatic digestion was successful. However, other GlcNAc binding lectins (WGA and STL), all of the GalNAc binding lectins (Moa, CFL, DBL and WFL), all of the sialic acid binding lectins (SNA, MAL-I, SSA) and the T-antigen binding lectins (ABL, ACA, BS-II) maintained the binding fluorescence signal at the native EV-glycosylation levels (Figure 4B).

## 3. Discussion

In this report, we describe the crucial role of N-glycosylation in MSC-EV. MSC-EV have been potentially explored as therapeutic agents, due to their appropriate characteristics [31]. They have already showed modulation of inflammatory pathologies in vivo in several animal models (reviewed in [1,32]) and also in human clinical trials (reviewed in [2]). However, the EV mechanism of action is still not fully elucidated. In addition, EV–cell interaction is still a challenge for the scientific community, and many proteins have been proposed to be involved, such as tetraspanins, integrins or lectins. Specifically, antibody-mediated blocking of CD9 and CD81 tetraspanins showed a decreased uptake of EV by dendritic cells [33]. Similarly, the antibodies against integrins or lectins, such as CD11a or DC-SIGN, showed the same effect [33,34]. In the last few years, the development of the tools and techniques allowing the characterization of the glycans’ structure, such as lectin glycoarrays [35], mass spectrometry [36], integrated magnetic analysis of glycans in extracellular vesicles (IMAGE; [37]) and prolonged ultracentrifugation with electrostatic repulsion–hydrophilic interaction chromatography (PUC-ERLIC; [38]), has allowed the thorough examination of the glycan structures, such as the abundant glycoconjugates (proteoglycans, glycoproteins) covering the EV surfaces [39]. These studies indicate that the glycosylation pattern of the molecules might be the responsible of the EV–cell interaction. Thus, in this study we studied the role of N-glycosylation in MSC-EV capture and functionality. We also characterized the glycan profile of the intact MSC-EV and PNGase-F-treated MSC-EV.

The removal of the MSC-EV N-glycosylation by PNGase-F treatment decreased the EV–cell interaction and uptake by HUVEC, indicating that MSC-EV N-glycosylation plays a crucial role in these recognition and internalization processes. These results are in line with the previous observations of inhibition of the capture of the helminth pathogen, *Fasciola hepatica*-derived EV, by macrophages after treatment with PNGase-F [40], and also altering the functions of the cancer-specific blood-derived microvesicles [41]. However, in the EV derived from hepatic cell lines [26] and the EV isolated from brain-metastatic BMD2a cells [42], the cleavage of N-glycans resulted in an increased uptake, indicating that EV-glycans have an EV-source specificity and might regulate interactions and tissue targeting. In fact, several studies have already shown that the modification of EV glycosylation can alter their biodistribution in mice models [28,42].

The ability of EV to stimulate cell migration has already been reported by several groups for different cell sources. The cancer cell-derived EV were shown to modulate the dissemination pattern of metastatic cells [43] and, in particular, MSC-EV have been seen to stimulate chondrocyte and synovial MSC migration via the CXCL5 and CXCL6/CXCR2 axes [12]. However, none of these studies referred to glycan-dependent mechanisms. As we previously demonstrated in a porcine model using progenitor endothelial cells [30], MSC-EV hold the ability to recruit endothelial cells. Additionally, here we show the important role of the iWJ-MSC-EV N-glycosylation in this crucial process. Only the agarose spots containing native EV had an increased ability to recruit HUVEC, while the migration induction was reduced by half in the spots containing PNGase-F-treated iWJ-MSC-EV. However, the most affected function in our setting was angiogenesis. In this regard, the tube-like formation by HUVEC was drastically inhibited in the PNGase-F-treated iWJ-MSC-EV. These different levels of affectation may indicate that N-glycans could be differently involved in these functions. The decreased number of nodes in the presence of the PNGase-F-treated EV evidenced the difficulty of HUVEC to establish cell-to-cell interactions, the importance of the protein N-glycosylation in promoting these contacts and the possible abrogation of the EV–HUVEC’s receptors interaction, responsible for angiogenesis induction. The mechanisms of MSC-EV to induce angiogenesis were related to the vascular endothelial growth factor (VEGF) pathway. Mouse MSC-EV were found to carry VEGF, and to promote the expression of VEGF receptors 1 and 2 in mouse endothelial cells, stimulating other pro-angiogenic pathways, such as SRC, AKT and ERK [44]. In this context and related to glycans, the heparan sulfate proteoglycans have shown to enhance the cell membrane docking sites for vascular endothelial growth factor (VEGF) [45]. In addition, N-glycans were found necessary for several ligands of some of the tyrosine kinase receptors (RTKs), such as VEGFR [46,47,48]. Altogether, these observations point to the activation of the VEGFR in a glycan-dependent manner that could explain the angiogenesis stimulation.

Given the implication of MSC-EV N-glycosylation in their uptake and function, we further analyzed the glycan profile of both of the types of MSC-EV in order to identify which were the glycan structures affected by the PNGase-F treatment. Our lectin glycoarray revealed that the intact MSC-EV were mainly enriched in GlcNAc and GalNAc; the fucosylated glycans and high-mannose glycans were less abundant, and the sialic acids and T-antigens were scarcely found. In contrast, other studies reporting MSC-EV glycosylation patterns found a completely different profile, especially of the sialic acids [25,49]. Furthermore, different patterns were reported in different sources of MSC [50]. These evidences support that the different sources of a same cell type or tissue may have their own glycan profiles. These outcomes, together with the glycan diversity reported of EV derived from other sources (cancer, urine and plasma), point to EV-source glycan specificity. However, and importantly, this diversity may be also attributed to the type of EV isolation method used. In fact, it has been demonstrated that the EV isolation strategy can directly impact on the final EV glycosylation pattern [51].

The PNGase-F-treated EV resulted in decreased amounts of the majority of the above-mentioned glycans when compared to wild-type EV. Additional evidence of the enzyme treatment was the reduction in all of the EV markers analyzed (i.e., CD9, CD63, CD90), indicating the presence of sensitive epitopes for PNGase-F. In fact, the antibody-binding alterations due to glycosylation were already reported [52,53]. The high diversity of the glycans affected by PNGase-F treatment precluded any identification of the specific targets involved in the analyzed functions. In this sense, mass spectrometry analysis may help to discover the molecular specifics of these glycans, distinguish between N-glycan and O-glycan associated to these residues and identify which ones are linked to the proteins.

In summary, our data show the crucial role of N-glycans in MSC-EV surface for their uptake and interaction with target HUVEC which, in turn, affected the cell functionality. The implication of the glycans in these crucial biological processes suggest that the modification of the glycan profiles in MSC-EV may be a possible way to alter the dynamics of EV–target cell interactions for forthcoming therapeutic uses.

## 4. Materials and Methods

All of the study protocols with human samples were approved by the Clinical Research Ethics Committee of our institution (Germans Trias i Pujol University Hospital; EO-10-016 and EO-12-022), and conformed to the principles outlined in the Declaration of Helsinki.

### 4.1. Cell Culture

The MSC were cultured in T-175 flask in complete α-Minimum Essential Medium (α-MEM; Sigma Aldrich, St. Louis, MO, USA) supplemented with 10% heat-inactivated fetal bovine serum (FBS; Life Technologies, Carlsbad, CA, USA), 2 mM L-glutamine and 1% of penicillin-streptomycin (P/S; Gibco Invitrogen Corp., Carlsbad, CA, USA); at 5% CO_2_ and 37 °C.

The Wharton’s jelly-derived MSC (WJ-MSC) were isolated as previously described [29]. Succinctly, with the parents’ consent, fresh umbilical cords were obtained after birth and maintained in phosphate-buffered saline buffer (PBS; Gibco Life Technologies/Invitrogen, Carlsbad, CA, USA) supplemented with 5000 U heparin (Sigma Aldrich, St. Louis, MO, USA) and 1% P/S. The umbilical cords were cut into 3–6 mm^3^ sections, washed with PBS and disrupted mechanically. Next, two enzymatic digestions were performed and both of the supernatants obtained were mixed together and centrifuged. The cell pellet was finally resuspended in complete α-MEM and MSC selected by plastic adhesion in culture.

In order to reduce the batch to batch variations, the WJ-MSC from three donors with confirmed immunosuppression capabilities [29] were immortalized by hTERT and SV40 transduction, following a previously published protocol [54], and referred here onwards as iWJ-MSC.

Briefly, the Phoenix AMPHO cells were seeded till 50% confluence and were transfected with hTERT and SV40 early region constructs, using the DNA and calcium phosphate precipitation method. The mix (DNA, CaCl_2_ solution, sterile water and 2 × HBS solution) was added to the cells and they were incubated overnight at 37 °C or 8 h at 37 °C in the presence of 10 µM chloroquine. The transfection medium was replaced with fresh medium. After 24 h, the cell supernatant containing retrovirus viral particles was collected, passed through a 0.45 µm filter and 1 µg/mL polybrene was added. Finally, the WJ-MSC, already in culture, were infected by replacing their medium with the filtered viral supernatant taken from the Phoenix AMPHO culture and centrifuged for 45 min at 750× *g* and at 32 °C. After centrifuging, the medium was removed and fresh α-MEM medium was added. The infection procedure was repeated twice at 24 h intervals. Finally, the iWJ-MSC cells were selected with 50 µg/mL Geneticin (G418) and 50 µg/mL of Hygromicin (Gibco Life Technologies/Invitrogen).

The human umbilical vein endothelial cells (HUVEC) were cultured in pre-coated 6-well plates with 0.05% of gelatin A (Sigma Aldrich), in Endothelial Growth Media (EGM-2) (Lonza, Basel, Switzerland); at 5% of CO_2_ and 37 °C.

### 4.2. EV Isolation

The EV were isolated from the cell culture supernatant by size-exclusion chromatography (SEC), following the previously published protocol [55]. First, the serum-derived bovine EV were depleted by ultracentrifugation of 2× complete medium in polyallomer ultracentrifugation tubes (Thermo Fisher Scientific, San Diego, CA, USA) at 100,000× *g* for more than 16 h (TH641 rotor, adjusted k-Factor = 24,082) in a Sorvall WX Ultra 100 Series ultracentrifuge (Thermo Fisher Scientific). The supernatant was then sterilized using a 0.22 μm filter (Starsted, Nümbrecht, Germany) and diluted with α-MEM to a 1× working solution for cell culture.

For the EV production, the MSC were cultured in a T-175 flask until confluence was reached (10^7^ cells approximately), then the medium was replaced with 15 mL 1× *g* ultracentrifuged medium. After 48 h, the conditioned medium (CM) was collected and sequentially centrifuged at 400× *g* for 5 min to exclude the cells and at 2000× *g* for 10 min to eliminate the cell debris. To obtain the concentrated CM (CCM), the CM was ultrafiltered at 2000× *g* for 40 min using a 100-kDa Amicon Ultra centrifugal filter (Millipore, Billerica, MA, USA). Then, the EV were isolated by SEC using 1 mL Sepharose CL-2B (Sigma Aldrich) packed on the SEC column, equilibrated with PBS. Then, 100 μL of CCM was loaded to the column, with sterile 1× PBS used as the elution buffer, and twenty 100 μL-fractions were collected.

Finally, the EV-containing fractions were determined by bead-based flow cytometry, and protein elution was checked by reading the absorbance at 280 nm in a Nanodrop spectrophotometer (Thermo Fisher Scientific). The EV-containing fractions were pooled together to be used in further experiments.

### 4.3. PNGase-F Treatment of iWJ-MSC-EV

The surface N-glycans of iWJ-MSC-EV were removed by incubation with the enzyme PNGase-F. Specifically, 10 U of PNGase-F (N-glycosidase F; Roche Diagnostics GmbH, Mannheim, Germany) were added to 100 µL of CCM and incubated for 18 h at 37 °C before EV isolation. The enzyme was eliminated by the downstream SEC purification of iWJ-MSC-EV.

### 4.4. EV Staining

The fluorescently-labelled iWJ-MSC-EV were produced for tracking purposes. Briefly, 100 μL of CCM was stained with 20 µM carboxyfluorescein succinimidyl ester (CFSE; eBioscience^TM^, Invitrogen) for 2 h at 37 °C [56] before EV isolation. The CFSE excess dye was washed by the downstream SEC purification of the labelled EV.

### 4.5. Bead-Based Flow Cytometry

The presence of EV in SEC fractions was detected using bead-based flow cytometry [55]. First, 20 μL of the SEC fractions were coupled to 4 μL of 4 μm aldehyde/sulphate-latex beads (Invitrogen-Thermo Fisher Scientific), incubated for 15 min at room temperature (RT), and then blocked in 1 mL BCB buffer (PBS, 0.1% bovine serum albumin (BSA), and 0.01% sodium azide (NaN_3_); Sigma Aldrich) for 2 h on rotation. Next, the EV-coated beads were centrifuged at 2000× *g* for 10 min, and resuspended in BCB buffer. The antibody labelling was performed by incubation (30 min at RT) with the fluorochrome-conjugated antibody anti-CD90-PE-Cy7 (1:50; BD, San Diego, CA, USA) or by indirect labelling with the primary antibodies anti-CD63 (1:10, Clone TEA3/18) and anti-CD9 (1:10, Clone VJ1/20.3.1), followed by incubation with the secondary antibody FITC-conjugated goat F(ab’)2 anti-mouse IgG (1:100; Bionova, Halifax, NS, Canada) or with the Alexa647-conjugated donkey F(ab’)2 anti-mouse IgG (1:100; Jackson ImmunoResearch Europe Ltd., Ely, UK) in CFSE-labelled EV. The beads were washed with the BCB buffer after each incubation by centrifugation at 2000× *g* for 10 min. Finally, the data were acquired in the FACSLyric flow cytometer (BD) and analyzed using FlowJo v10.2 software (FlowJo, LLC, Ashland, OR, USA).

### 4.6. Dotblot

The negative control marker calnexin and the EV-positive marker ezrin were evaluated by dotblot. Briefly, 2 µL of EV sample or a cell lysate as control were spot onto a nitrocellulose membrane (Whatman Protran, Dassel, Germany). Next, the non-specific sites of the membrane were blocked in BSA/TBS-T (0.1% BSA in 20 mM Tris-HCl, 150 mM NaCl, pH 7.5 + 0.05% Tween20) for 30 min at RT. Incubation with the primary antibodies calnexin (1:250, SC-23954; Santa Cruz Biotechnology, Inc., Dallas, TX, USA) and ezrin (1:5000, ab40839; Abcam, Cambridge, UK) was performed for 1 h at RT and on rotation, followed by an incubation with the secondary antibodies IRDye800-conjugated goat F(ab’)2 anti-mouse IgG (1:15,000; LI-COR Biosciences, Lincoln, NE, USA) and IRDye680-conjugated goat F(ab’)2 anti-rabbit IgG (1:15,000; LI-COR Biosciences, Lincoln, NE, USA). The membrane was washed with TBS-T for 10 min on rotation after each incubation. Finally, the membrane was scanned in an Odyssey CLx Scanner (LI-COR Biosciences, Lincoln, NE, USA).

### 4.7. Cryo-Electron Microscopy

The EV size and morphology were analyzed in vitrified EV pools by cryo-electron microscopy (cryo-EM), using a transmission electron microscope (Joel JEM 2011, Tokyo, Japan) as previously described [29].

### 4.8. iWJ-MSC-EV Capture Assays

The capture of the iWJ-MSC-EV was first assessed in HUVEC by flow cytometry. A total of 50,000 HUVEC were seeded in 96-well plates pre-coated with 0.05% gelatin A (Sigma Aldrich, St. Louis, MO, USA), incubated for 24 h to ensure adhesion and stabilization. The CFSE-labelled iWJ-MSC-EV (from 5 × 10^5^ producing cells), pre-treated or not with PNGase-F, were added. After 1 h incubation, the cells were washed twice with 100 µL PBS and trypsinized. The cells were resuspended in 100 µL of EGM-2 and acquired in a Canto II Flow Cytometer (BD). Finally, the EV capture was analyzed, using the FlowJo software (version 10.6.2) as the percentage of CFSE^+^ cells. The fluorescence microscopy was also performed to confirm the flow cytometry results. In this case, 50,000 HUVEC were seeded in each channel of a 0.05% gelatin pre-coated µ-slide VI 0.4 (Ibidi, Gräfelfing, Germany) and incubated overnight to ensure adhesion. As before, the CFSE-labelled iWJ-MSC-EV, treated or not with the PNGase-F, were added. After 1 h incubation, the cells were washed four times with 100 µL PBS and the fluorescence images were captured using an Axio-Observer Z1 confocal microscope (Zeiss, Oberkochen, Germany). The EV capture was analyzed taking 40× oil magnification, capturing red (autofluorescence control) and green (CFSE-EV) fluorescence signals and the phase contrast used to determine the cell edges and number. The quantification of the fluorescence was performed using Fiji software (ImageJ, NIH). As the CFSE-EV fluorescence was very dim, the threshold from the green fluorescence was lowered, which in turn increased the autofluorescence of the HUVEC cells. Thus, to distinguish the autofluorescence from specific CFSE-EV fluorescence, the green fluorescence that co-localized with the red fluorescence was considered autofluorescence, while the green fluorescence with no co-localization was counted as the specific EV fluorescence (EV capture). Briefly, to obtain the specific CFSE-EV green fluorescence, the mean of the green fluorescence (G) of each picture was normalized by the means of the red fluorescence (R). If the G/R ratio was lower than the mean G/R ratio of the controls (HUVEC alone), the green area and the mean green fluorescence was considered 0. All of the signal above the mean G/R ratio of the controls was considered specific green CFSE-EV fluorescence.

### 4.9. Agarose Spot Migration Assay

The cell migration was measured in agarose spots using an adaptation of a previously described protocol [57]. First, the wells of a 6-well plate were divided into four drawn quadrants and pre-coated with 10 µg/mL of fibronectin (Merck Millipore, Darmstadt, Germany) to ensure HUVEC adhesion to plastic. Then, three spots of 5 μL of 0.5% low-melting point agarose (40 °C, Ecogen, Barcelona, Spain) mixed 1:1 with either PBS (used as buffer) or iWJ-MSC-EV, treated or not with PNGase-F (from 2.5 × 10^5^ producing cells), were placed on the surface of a different quadrant. Next, the plate was incubated at 4 °C for 15 min to ensure agarose jellification and adherence. Then, 1 × 10^5^ HUVEC were seeded over each well of EGM-2 medium. The cells were incubated for 24 h at 37 °C to allow the cells’ invasion of the solidified agarose drops. After incubation, four images from every firmly attached agarose spot of each condition were taken with a phase contrast inverted microscope at 10× objective magnifications (CKX41, Olympus Life Science, Center Valley, PA, USA), and migration was measured using the Fiji software (ImageJ, NIH). Specifically, both the distance of the three most distant cells from the agarose spot limit were measured for each spot, as well as the area of the spot occupied by the migrating cells, measured using the publicly available macro tool (https://imagej.nih.gov/ij/macros/SA_NJ.txt) (accessed on 26 September 2019).

### 4.10. Matrigel-Based Tube Formation Assay

The tube-like formation capacity was tested using HUVEC in a 24-well plate. The wells were pre-coated with 200 µL of Matrigel (Corning, NY, USA), incubated for 10 min at RT and 30 min at 37 °C. Then, 15,000 HUVEC/well were seeded, and cultured in complete EGM-2 without VEGF with iWJ-MSC-EV from 2.5 × 10^5^ or 5 × 10^5^ producing cells, treated or not with PNGase-F. The VEGF-A (10 ng/mL, Sigma Aldrich, St. Louis, MO, USA) was used as a positive control. The cells were incubated for 6 h and images of each well were taken and analyzed using the Image J–Angiogenesis analyzer tool (ImageJ, NIH).

### 4.11. Glycoarray

The lectin microarray analysis of EV was adapted from the previously published methodology [26,58]. The arrays were fabricated using NHS-functionalized glass slides (NEXTERION Slide H, SCHOTT, Jena, Germany) and 0.5 mg/mL solutions of lectins reconstituted in a printing buffer of PBS with 0.5% glycerol and 0.005% Tween-20. A sciFLEXARRAYER S11 noncontact piezoelectric spotter (Scienion, Berlin, Germany) was employed to print the six replicates spots of 0.67 nL for each lectin, with spacing of 0.30 mm spaces between the spots. The lectin-spotted slides were left to react and dry overnight in the printing chamber at a relative humidity of 50% and 18 °C temperature, before vacuum sealing and storage at −20 °C.

For the analysis, the EV were diluted to 50 μg/mL in PBS and labelled using 60 μg/mL Alexa Fluor 647 NHS ester (Invitrogen, Thermo Fisher Scientific) for 1 h at RT. Then, 1M Tris buffer was used to quench the unreacted NHS dye for a further hour at RT. For the incubation with the arrays, the labelled EV were diluted to 5 μg/mL in PBS containing 0.3 mg/mL BSA, 0.1 mM CaCl_2_, 0.1 mM MgCl_2_ and 0.005% Tween-20 (‘incubation buffer’). The microarray slides were retrieved from −20 °C storage on the day of use and the unreacted NHS groups were quenched with 30 mM ethanolamine in a borate buffer for 1 h at RT, before equilibration with an incubation buffer for a further hour. The EV were then incubated with arrays for 3 h at RT and then washed twice with PBS, dried by centrifugation and scanned with an Agilent G2565BA microarray scanner (Agilent Technologies, Santa Clara, CA, USA) at photomultiplier tube voltage set to 20% of the maximum. The fluorescence intensities for all of the spots were extracted using ProScanArray Express v4.0 software (PerkinElmer, Waltham, MA, USA). The subsequent data processing was performed by normalizing the fluorescence value of each spot to the highest binding signal within each array before the combining of spot replicates for the average values and standard deviations.

### 4.12. Statistics

The data are shown as mean (SD). The normality of the data was checked by a Shapiro–Wilk test in all of the datasets and the appropriate statistical test for each case was performed. Only significant statistical differences (*p* < 0.05) are shown. The analyses were performed using GraphPad Prism software (9.0 version).

## Figures and Tables

**Figure 1 ijms-23-09539-f001:**
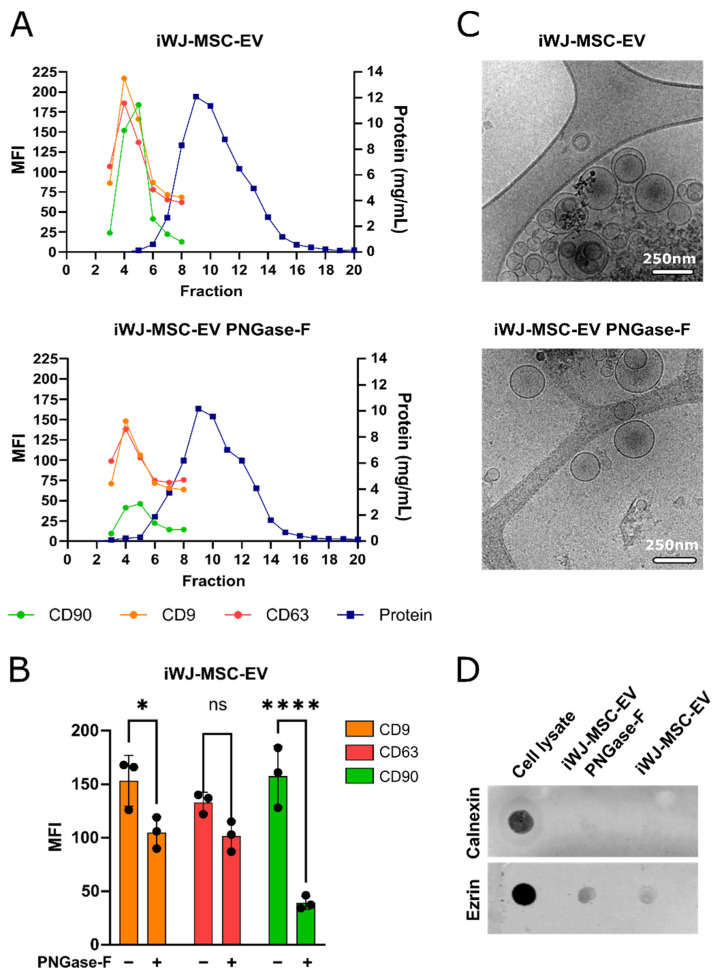
EV were isolated from iWJ-MSC conditioned medium by SEC and characterized. (**A**) Representative SEC elution profile of iWJ-MSC-EV, treated or not with PNGase-F, obtained by bead-based flow cytometry and positive for the EV and MSC markers CD63, CD9 and CD90. Protein elution was measured at 280nm absorbance and occurred later; (**B**) Comparative analysis of MSC-EV markers between untreated and PNGase-F-treated iWJ-MSC-EV. Each dot corresponds to independent EV characterizations. The statistic differences are indicated for * *p* < 0.05 and **** *p* < 0.0001 by two-way ANOVA with a Šídák’s multiple comparisons test; (**C**) Representative images of untreated and PNGase-F-treated iWJ-MSC-EV taken by cryo-transmission electron microscopy (cryo-TEM). iWJ-MSC-EV were double membraned nanovesicles of 50–500 nm, independent of treatment. Scale bars are 250 nm; (**D**) Calnexin and ezrin dotblots of both types of iWJ-MSC-EV. PBMC cell lysate is used as positive control.

**Figure 2 ijms-23-09539-f002:**
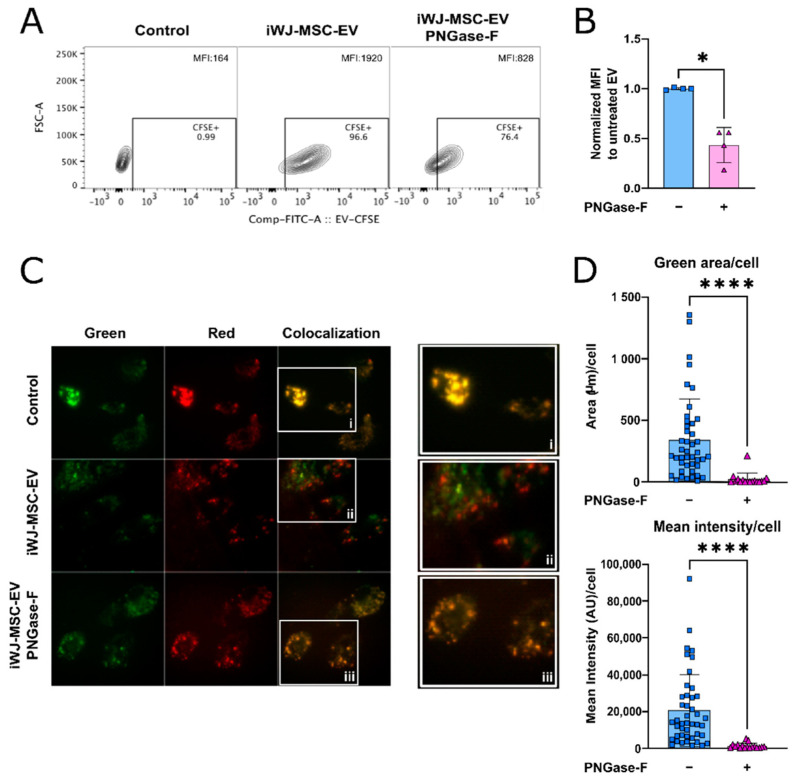
Capture of untreated and PNGase-F-treated iWJ-MSC-EV by HUVEC, after 1h of incubation. (**A**) Representative flow cytometry analysis of EV uptake by HUVEC, represented in counter plots (% of CFSE^+^ cells); (**B**) Quantification of HUVEC’s MFI by flow cytometry. Three biological replicates are represented with one or two experimental replicates; (**C**) Fluorescence microscopy uptake analysis. Representative images of HUVEC with PBS (control), untreated iWJ-MSC-EV-CFSE and PNGase-F-treated iWJ-MSC-EV-CFSE. Green signal corresponds to CFSE-labelled EV and the Red channel was used to control the signal attributable to autofluorescence as seen by co-localization of both (right). Images were taken at 40× (oil objective); (**D**) Quantification of the EV uptake according to green area/cell and the mean intensity/cell after G/R ratio correction of the data. Data from two independent experiments with a minimum of 15 images/condition. The statistic differences are indicated for * *p* < 0.05 and **** *p* < 0.0001 by a Kruskal–Wallis test with a Dunn’s post-hoc analysis.

**Figure 3 ijms-23-09539-f003:**
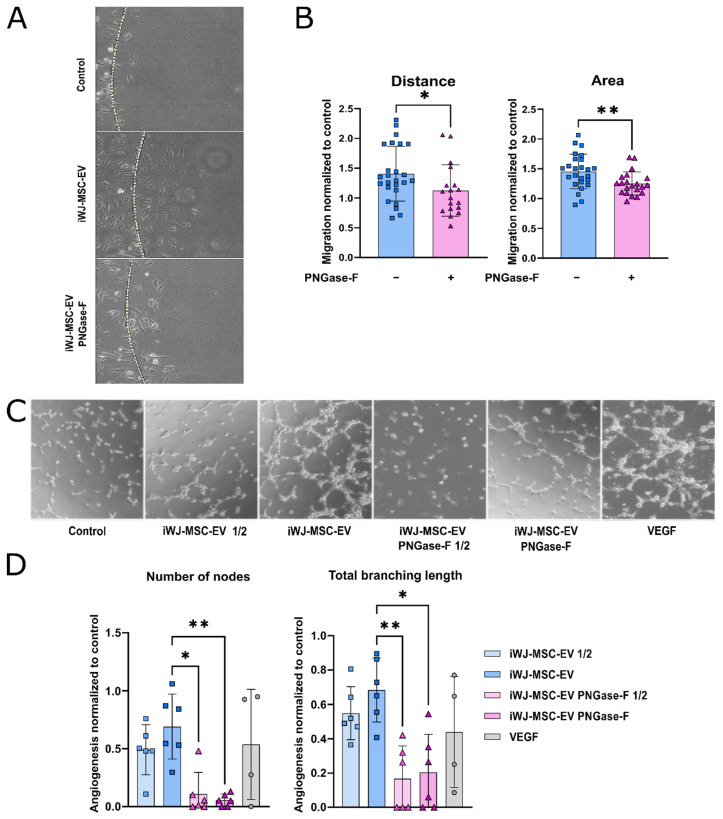
N-glycosylation is important for recruitment and pro-angiogenic functions of iWJ-MSC-EV. (**A**,**B**) Recruitment of HUVEC by iWJ-MSC in agarose spot assay; (**A**) Representative images (10×) of the HUVEC’s migration from all the conditions. Agarose spot’s border is represented as a dotted line; (**B**). Quantification of HUVEC’s migration. The distance of the three most distant cells from the spot’s limit from each image was quantified. Also, the area occupied by the cells inside the agarose spot; (**C**) Representative images (10×) of the angiogenesis reached by all the conditions; (**D**) Quantification of the angiogenesis as number of nodes formed and total branching length of the tube formation. Data from two independent experiments with a minimum of 3 replicates/condition. The statistic differences are indicated for * *p* < 0.05 and ** *p* < 0.01, by a Kruskal–Wallis test with a Dunn’s post-hoc analysis.

**Figure 4 ijms-23-09539-f004:**
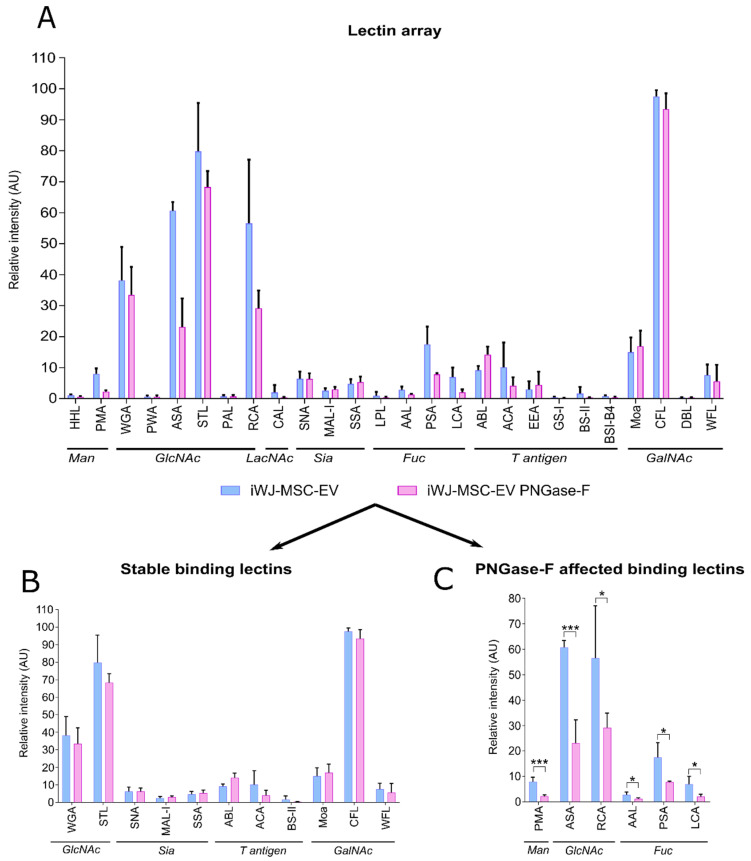
(**A**) Complete lectin array analysis of iWJ-MSC-EV and PNGase-F-treated iWJ-MSC-EV; (**B**) Representation of the binding lectins in which binding signal remained stable after PNGase-F treatment; (**C**) Representation of the binding lectins affected by the PNGase-F treatment. Data from two biological replicates and 6 different technical replicates per lectin. The statistic differences are indicated for * *p* < 0.05 and *** *p* < 0.001 by a multiple *t*-test. *Man*: *mannose*; *GlcNAc*: *N-acetylglucosamine*; *Fuc*: *fucose*; *Sia*: *sialic acid*; *GalNAc*: *N-acetylgalactosamine*.

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
