# Peer review of "N-Glycans in Immortalized Mesenchymal Stromal Cell-Derived Extracellular Vesicles Are Critical for EV–Cell Interaction and Functional Activation of Endothelial Cells"

_ijms, 2022, doi:10.3390/ijms23179539_

Round 1

Reviewer 1 Report

This is a nice work with good set of experiments. I have few comments to improve the manuscript.

1. For EV production, MSCs were cultured in a T-175 flask until confluence. Is it a good practice to keep the cells till confluence or ~80% would be ideal? May be at 100% confluence for 2 days in CM  the quality of EVs may not be ideal for experiments. 

2. The proangiogenic effects of MSC-EVs at 6 hours looks drastic compared to controls. Do you think they have high levels of VEGF or any other factors are responsible? Please reconfirm the data. 

Author Response

Reviewer #1: This is a nice work with good set of experiments. I have few comments to improve the manuscript.

We thank the reviewer for the comments.

  1. For EV production, MSCs were cultured in a T-175 flask until confluence. Is it a good practice to keep the cells till confluence or ~80% would be ideal? May be at 100% confluence for 2 days in CM the quality of EVs may not be ideal for experiments. 

We agree with the reviewer that culture conditions can impact directly on the EV produced. For this reason, we have checked the functionality of the produced EV in different starting cell confluence (60-100%) and functionality (T cell proliferation inhibition) did not change due to cells’ confluence, which has already been observed by other groups, such as by Patel et al., 2017. Thus, EV produced by confluent cells are also suitable and functional for experiments. https://pubmed.ncbi.nlm.nih.gov/28932818/

  1. The proangiogenic effects of MSC-EVs at 6 hours looks drastic compared to controls. Do you think they have high levels of VEGF or any other factors are responsible? Please reconfirm the data

The data clearly shows the high pro-angiogenic potential of MSC-EV as the reviewer points out. We performed the experiments with three different iWJ-MSC and could repeatedly obtain a high proangiogenic effect compared to controls.

Following the reviewer’s suggestion, we have tested the presence of VEGF-D in our iWJ-MSC-EV and iWJ-MSC-EV treated with PNGase-F and levels were under the limit of detection (<1,6 pg/mL) in all samples. Nevertheless, due to time constraints, we could not have access to VEGF-A detection assays. Of note, when we performed a previous proteomic study of non-immortalized WJ-MSC-EV, we could not detect any VEGF family isoforms (Cabrera-Pérez et al., 2019). However, other studies using MSC-EV have detected VEGF by qRT-PCR and have associated their levels to the angiogenic functionality of the MSC-EV, as VEGF receptors were increased in target cells (Gangadaran et al., 2017).

References:

Cabrera-Pérez et al., 2019: https://pubmed.ncbi.nlm.nih.gov/31779673/

Gangadaran et al., 2017: https://pubmed.ncbi.nlm.nih.gov/28837823/

Reviewer 2 Report

Overall, the manuscript is sound and experiment approaches and results were well presented. Some minor revisions would further strengthen the manuscript.

There is a lack of description in the manuscript on the sub-cellular localization of the staining in Fig 2C. Also, it seems like there are some nuclear staining in the control group. It would be great to add DAPI/ nuclear staining to clearly show the sub-cellular localization. 

Most of the experiments in this study used a single approach with single cell line. It would be great to validate the findings with orthogonal methods and different cell line(s). For example, angiogenesis assay in Fig 3 can be validated with bovine aortic ring angiogenesis assay

Validation on the results from the lectin array is required (at least a couple of the significant ones, if possible)

Author Response

A formatted version of the point-by-point answers to the reviewers comments has been uploaded as a pdf file.

Reviewer #2: Overall, the manuscript is sound and experiment approaches and results were well presented. Some minor revisions would further strengthen the manuscript.

We thank the reviewer for the comments and recommendations.

  1. There is a lack of description in the manuscript on the sub-cellular localization of the staining in Fig 2C. Also, it seems like there are some nuclear staining in the control group. It would be great to add DAPI/ nuclear staining to clearly show the sub-cellular localization. 

We agree with the reviewer that there is no specific analysis in the manuscript about the sub-cellular localization of the EV capture. We wanted to clarify first the nuclear staining observed in the control group, which is due to cells’ autofluorescence. As explained in M&M, when the same nuclear staining is found in both channels (green and red) it can be attributed to autofluorescence only. A similar phenomenon can be clearly seen in the PNGase-F treated iWJ-MSC-EV condition, but distinct green-only signal was observed when intact iWJ-MSC-EV are added. Captured EV signal in target cells can be clearly observed as a specific EV-derived green fluorescence (lacking red signal), and they were repeatedly localized in the cytoplasm.

Following the reviewer’s recommendation, we checked the EV subcellular localization in cells stained with DAPI to delineate the nucleus limits, and we could corroborate the cytoplasm localization of the EV. Nevertheless, a high spillover signal was then found in the green channel after nucleus staining, thus we did not consider this option to be optimal to detect and clearly differentiate the dim EV signal from the increased background. We attach here the images for the reviewer to evaluate (Figure R1).

Accordingly, we added in line149 a comment related to this:

“Our results showed specific green fluorescence (CFSE-EV) in HUVEC’s cytoplasm after the capture of untreated EV. After quantification, untreated EV uptake by HUVEC was significant in terms of mean intensity per cell and green area per cell after autofluorescence correction.”

Figure R1: Capture of untreated iWJ-MSC-EV by HUVEC, after 1h of incubation. Fluorescence microscopy uptake analysis. Representative images of HUVEC with PBS (control) or untreated iWJ-MSC-EV-CFSE. HUVEC’s nuclei were stained with DAPI (blue signal, left). A high spillover of DAPI can be observed in the green channel (middle). Green signal with no co-localization with blue fluorescence corresponded to specific CFSE-labelled EV signal (down, right). Images were taken at 40x (oil objective).  

  1. Most of the experiments in this study used a single approach with single cell line. It would be great to validate the findings with orthogonal methods and different cell line(s). For example, angiogenesis assay in Fig 3 can be validated with bovine aortic ring angiogenesis assay

We thank the reviewer for the recommendations. Related to using other cell lines, we have also performed the agarose spot migration assay with allogeneic subcutaneous fat MSC in the context of another scientific study, and we can ensure that WJ-MSC-EV are able also to recruit MSC to the same extent as HUVEC. However, we have not shown the results in this paper, as we wanted to focus on human endothelial cells as an important MSC-EV target in ischemia-reperfusion injuries. Thus, all the experiments were only performed with HUVEC.

Furthermore, in a previous published study from our group, we showed the translation of this same properties of WJ-MSC-EV in the swine species. Porcine adipose tissue MSC-EV were able to recruit both MSC and progenitor outgrowth endothelial cells (OEC) in the agarose spot migration assay, and induced the tube-like formation to outgrowth endothelial cells (OEC) with similar results as human tissue-derived MSC (Monguió-Tortajada et al,. 2021). Therefore, we are confident the results presented here are robust and reproducible.

Validating the angiogenesis results using another approach would be great, however, we do not have access to bovine aortic rings as suggested by the reviewer. Furthermore, we focused our experiments in using human cells to do the functional assays to better fit our objective, and use allogeneic setting instead of xenogeneic (as would be using bovine endothelial cells). We think that the tube-like formation assay performed is already well stablished in the scientific community and results are reproducible and robust, as we validated in both human and porcine settings and even using other tissue-derived MSC and endothelial cell sources.

References:

Monguió-Tortajada et al., 2021: https://www.ncbi.nlm.nih.gov/pmc/articles/PMC7973387/

  1. Validation on the results from the lectin array is required (at least a couple of the significant ones, if possible)

To validate the lectin array, we have repeated it using iWJ-MSC-EV derived from another iWJ-MSC cell line. The glycan profile observed was very similar, confirming that the iWJ-MSC lines are comparable and that the PNGase-F enzyme is acting the same way, cutting the N-glycosylation of the same residue types (mainly GlcNAc). Furthermore, the very few changes observed adding a second validation sample made results more robust:

  • We observed mild change in WFL (GalNAc-binding lectin group), whose low-binding signal was no longer significantly affected by the PNGase-F enzyme treatment. This correlates now with the rest of lectins belonging to the GalNAc-binding lectins, showing no signal reduction after PNGase-F treatment.
  • The AAL lectin (Fucose-binding lectin group), reached now a significant reduction after PNGase-F treatment, analogously to PSA and LCA lectins of the same lectin group.

Accordingly, figure 4 has been updated with both cell lines’ analysis and the Abstract and Results section includes this new data:

Abstract: Line 64 “Finally, comparative lectin arrays of iWJ-MSC-EV and PNGase-F-treated iWJ-MSC-EV found marked differences in the surface glycosylation pattern, particularly in N-acetylglucosamine, mannose, and fucose binding lectins.”

Results: Line 221: “Lower binding was observed for fucose-binding lectins Pisum sativum (PSA) and Lens culinaris (LCA); the mannose-binding lectin Polygonatum multiflorum (PMA); and a very low presence of glycans containing sialic acids or O-glycans (T-antigen). Glycans containing α-galactose residues were absent (Figure 4A). In contrast, the PNGase-F-treated EV showed a decreased binding signal for some of these lectins. In particular, a reduction was observed in the GlcNAc-binding lectins (ASA and RCA), in the mannose-binding lectin (PMA) and in fucose-binding lectins (AAL, PSA and LCA) (Figure 4C). This strong decrease of the signal after PNGase-F treatment suggests that most of these glycosidic residues reside on N-glycans and that the enzymatic digestion was successful. However, other GlcNAc binding lectins (WGA and STL), all of the GalNAc binding lectins (Moa, CFL, DBL and WFL), all sialic acid binding lectins (SNA, MAL-I, SSA) and T-antigen binding lectins (ABL, ACA, BS-II) maintained the binding fluorescence signal at the native EV-glycosylation levels (Figure4B).”

Figure 4 caption: Line 242:” Data from two biological replicates and 6 different technical replicates per lectin.”

As discussed in the manuscript, a more exhaustive validation of the lectin array would require expensive and highly specialized techniques, such as dedicated mass spectrometry. We cannot validate specific glycan from the lectin array, as the lectin array can inform of the glycan types present in the EV but not the glycan structure or molecules to which they are attached to (proteins, lipids). Next steps would implicate to find highly glycosylated surface EV-proteins that contain the type of glycan that are affected by the PNGase-F treatment and evaluate their implication in the interaction and in the functional activities of iWJ-MSC-EV on target cells.
